# Highly Efficient Self-Assembled Activated Carbon Cloth-Templated Photocatalyst for NADH Regeneration and Photocatalytic Reduction of 4-Nitro Benzyl Alcohol

Vaibhav Gupta [1], Rajesh K. Yadav [1,*], Ahmad Umar [2,3,4,*,†], Ahmed A. Ibrahim [2,3], Satyam Singh [1], Rehana Shahin [1], Ravindra K. Shukla [1], Dhanesh Tiwary [5], Dilip Kumar Dwivedi [6], Alok Kumar Singh [7], Atresh Kumar Singh [7] and Sotirios Baskoutas [8]

1. Department of Chemistry and Environmental Science, Madan Mohan Malviya University of Technology, Gorakhpur 273010, India
2. Department of Chemistry, Faculty of Science and Arts, Promising Centre for Sensors and Electronic Devices (PCSED), Najran University, Najran 11001, Saudi Arabia
3. Centre for Scientific and Engineering Research, Najran University, Najran 11001, Saudi Arabia
4. Department of Materials Science and Engineering, The Ohio State University, Columbus, OH 43210, USA
5. Department of Chemistry, Indian Institute of Technology (BHU), Varanasi 221005, India
6. Department of Physics and Materials Science, Madan Mohan Malaviya University of Technology, Gorakhpur 273010, India
7. Department of Chemistry, Deen Dayal Upadhyaya Gorakhpur University, Gorakhpur 273009, India
8. Department of Materials Science, University of Patras, 26504 Patras, Greece
* Correspondence: rajeshkr_yadav2003@yahoo.co.in (R.K.Y.); ahmadumar786@gmail.com (A.U.)
† Adjunct Professor at the Department of Materials Science and Engineering, The Ohio State University, Columbus, OH 43210, USA.

**Abstract:** This manuscript emphasizes how structural assembling can facilitate the generation of solar chemicals and the synthesis of fine chemicals under solar light, which is a challenging task via a photocatalytic pathway. Solar energy utilization for pollution prevention through the reduction of organic chemicals is one of the most challenging tasks. In this field, a metal-based photocatalyst is an optional technique but has some drawbacks, such as low efficiency, a toxic nature, poor yield of photocatalytic products, and it is expensive. A metal-free activated carbon cloth (ACC)–templated photocatalyst is an alternative path to minimize these drawbacks. Herein, we design the synthesis and development of a metal-free self-assembled eriochrome cyanine R (EC-R) based ACC photocatalyst (EC-R@ACC), which has a higher molar extinction coefficient and an appropriate optical band gap in the visible region. The EC-R@ACC photocatalyst functions in a highly effective manner for the photocatalytic reduction of 4-nitro benzyl alcohol (4-NBA) into 4-amino benzyl alcohol (4-ABA) with a yield of 96% in 12 h. The synthesized EC-R@ACC photocatalyst also regenerates reduced forms of nicotinamide adenine dinucleotide (NADH) cofactor with a yield of 76.9% in 2 h. The calculated turnover number (TON) of the EC-R@ACC photocatalyst for the reduction of 4-nitrobenzyl alcohol is $1.769 \times 10^{19}$ molecules. The present research sets a new benchmark example in the area of organic transformation and artificial photocatalysis.

**Keywords:** EC-R@ACC photocatalyst; NADH regeneration; 4-nitro benzyl alcohol; solar light; photocatalytic

## 1. Introduction

Solar light has emerged as a sustainable and greener energy source for various solar chemical synthesis reactions. In the past few years, solar light-induced chemical transformations have been extensively achieved by eco-friendly processes [1,2]. In this context, the enlargement of an artificial substitute for this smart system continues to be an extraordinary challenge in the chemical society [1–6]. The recent research, therefore, involves synthesizing

and designing a photoreactor system as a photocatalyst for the selective regeneration of fine chemicals and the reduction of aromatic compounds under solar light. As reported previously, we noted that about 4% and 46% of the overall solar light accessible on the planet falls in the UV and visible ranges, respectively [7,8]. Consequently, a solar light i.e., a visible light-responsive photocatalyst is significantly important for the synthesis of solar fine chemicals, such as nicotinamide adenine dinucleotide phosphate (NADPH) and nicotinamide adenine dinucleotide (NADH) cofactor and the photoreduction of organic compounds. Additionally, a significantly important feature to be noted is that in photocatalytic systems, NADH is important for the fixation of carbon dioxide ($CO_2$). Therefore, the regeneration of the NADH cofactor and the reduction of the organic compound via a highly efficient pathway is the only way to make it economically and industrially feasible [9–12].

In this context, green chemistry and a related photoreactor, solar light, as one of the reactants of chemical synthesis, is a rising research area [13,14]. Solar light sponsored NADPH, NADH cofactor, and the photoreduction of organic compounds by various solar light harvestings materials, such as graphitic carbon nitride, graphene composites, and titanium oxide $TiO_2$, were well discovered [15,16]. Besides the photoreduction and degradation in the presence of solar light, different chemical reactions, such as [2 + 2] cycloaddition [17,18], reductive dehalogenation [19], hydrogen formation from 1,4 asymmetric alkylation [20,21], and dihydropyridine [22], are also described in the reported literature where various solar light-responsive metal complexes are utilized as the solar light-assisted photocatalyst. In spite of the utmost expensive metal complexes, solar chemical conversion can also be attained by using energy harvesting materials as a solar light-responsive photocatalyst [23]. It is evident from the literature that photocatalytic NADH regeneration and organic transformation consuming solar light illumination are a rising research zone, which provides various possibilities for future work. In this addition, $NAD^+$ is a bio-enzyme that needs a steady flux of solar light to photocatalyze a solar chemical synthesis [24]. Thus, solar light is essential in the chemical transformation reaction catalyzed by the enzyme. In the nonexistence of solar light and a photocatalyst, the $NAD^+$ type enzyme remains completely inactive during the catalytic reaction. To date, various types of photo enzymes have been investigated, which are used as a photosystem for organic transformation and solar chemical regeneration [25]. It is supposed that most of the possible formerly existing solar light active enzyme derivatives were sorted out by progression and that nowadays, solar light active coenzymes are only the past survivors of this pathway [24]. The use of many metal-based compounds for solar chemical regeneration and environmental remediation has achieved the utmost attention in current times due to the active utilization of naturally existing solar energy and an effective solar light active system to terminate various types of unwanted materials [26,27]. Furthermore, photosynthetic pathways can increase the fast and complete conversion of solar energy into solar fine chemicals [28]. Expensive metals such as CdS, ZnO, and $TiO_2$ are expensive materials utilized as a photocatalyst in various fields due to their good photocatalytic ability [29]. However, the key weakness of such types of photocatalysts is captured only in the ultraviolet (UV) quota of the solar light spectrum [30]. A diversity of semiconductor materials has been broadly utilized as light-harvesting photocatalysts by engineering or tuning the energy gap position and for effective use in the solar light spectrum [31]. Over the decades, expensive metal-based semiconductor materials have played an important role in solar light photocatalysis due to their narrow band gap and ionic conductivity, etc. [32,33]. The metal orbital along with a lone pair is combined with supporting materials, such as graphene, carbon activated cloth, and graphitic carbon nitride to generate a shift valence band (VB) and conduction band (CB) that tends to create the suitable band gap [34]. Among the metal-based photocatalysts, graphene has lately been utilized for the photocatalytic NADH/NADPH regeneration and conversion of organic substrates in polar and non-polar solvents under solar light illumination [35]. Expensive metals may exist as different crystalline phases along with scheelite tetragonal zircon tetragonal and scheelite-monoclinic [36]. It is a fact that the properties of the photocatalyst always depend on the nature of the crystal structure [37].

The monoclinic structure of a few expensive metals exhibits stronger photocatalyst ability under ultraviolet light illumination due to a small energy gap compared to other phasic structures [38]. Additionally, pure expensive metal has some restrictions, such as the rate of low absorption capacity of incident light, the fast recombination ability of solar light-created electrons, and a lack of pores in the structure [39]. It is of utmost importance to strengthen its solar light or solar spectrum absorption range ability and confine the recombination of solar light-generated holes and electrons in order to improve the ability of the photocatalyst under solar light illumination [40]. In this regard, many approaches have been designed by different researchers [16], such as n and p-type doping, fabricating a heterostructure solar light active system, and combining bias energy [41]. The combination of a few expensive metals with different types of porous materials, such as silica, alumina, glass, zeolites, and activated carbon cloth (ACC) is conducted to enhance the performance of the solar light absorption ability and overwhelm the charge carrier's recombination rate [42,43]. Among these, activated carbon cloth (ACC) has excellent physical and chemical properties to construct solar light active materials. The porous properties, structural stability, strong solar spectrum adsorption efficiency, and larger surface area of ACC enable healthier adsorption of substrates, making it a promising material that supports a photocatalytic procedure [44,45]. Thus, it permits the photocatalyst to absorb a solar spectrum that further leads to solar fine chemical production and the conversion of organic substrates [46,47]. A number of alterations have been described by many researchers on the solar light active catalyst surface using different supporting materials by various pathways of synthesis, such as sol-gel, co-precipitation, hydrothermal, solvothermal, and microwave synthesis [48]. Among these, the hydrothermal pathway is very easy to use to prepare different types of light-harvesting composites for solar chemical regeneration and organic transformations.

The solar chemical regeneration and photoreduction of NADH and organic compounds using solar radiation are developing new disciplines for future research. For the synthesis of new organic chemicals, the mechanism for the reduction of organic functionality is critical. To reduce organic functional groups, numerous methods have been reported, including (i) metal/acid reduction, (ii) photocatalytic reduction, (iii) electrolytic reduction, and (iv) catalytic hydrogen transfer [49]. One of the most prominent pollution control and disposal processes is the photocatalytic reduction of aromatic nitro compounds. Nitrogen-containing compounds are commonly created as by-products in a variety of industries and factories, including agrichemicals and pharmaceuticals. Among the many nitrogen-containing compounds, 4-nitrobenzyl alcohol is one of the most common by-products that is harmful to the environment [50,51].

4-amino benzyl alcohol (4-ABA) is made in the pharmaceutical industry by reducing 4-nitro benzyl alcohol (4-NBA). 4-ABA is a necessary precursor for the manufacture of a variety of drugs, including paracetamol, phenacetin, and acetanilide [52]. Metal-based photocatalysts in acidic conditions are utilized to reduce 4-NBA to 4-ABA to the greatest extent possible. However, such a procedure produces toxic metal oxide sludge that is harmful to the environment [52].

To address the aforementioned difficulties, a self-assembled metal-free self-assembled eriochrome cyanine R (EC-R) based ACC photocatalyst (EC-R@ACC) for the regeneration of NADH cofactors and the conversion of 4-NBA to 4-ABA is created. The synthesized metal-free EC-r@ACC photocatalyst has received a lot of interest in photocatalytic reactions because of its outstanding physicochemical properties, such as a suitable band gap, high molar extinction coefficient, low rate of intersystem crossing, excellent photocatalytic ability, easier synthesis, and excellent chemical stability. When compared to the metal-based photocatalyst, the metal-free self-assembled EC-r@ACC composite demonstrated significantly higher efficiency for NADH cofactor regeneration and the production of 4-ABA via artificial photocatalysis. Due to the utilization of environmentally acceptable and sustainable solar energy, artificial photocatalysis has sparked great interest in the synthesis of solar chemicals (NADH) and the reduction of 4-NBA to 4-ABA. A schematic

representation of the photocatalytic reduction of 4-NBA to 4-ABA and NADH regeneration under a solar light spectrum is represented in Scheme 1.

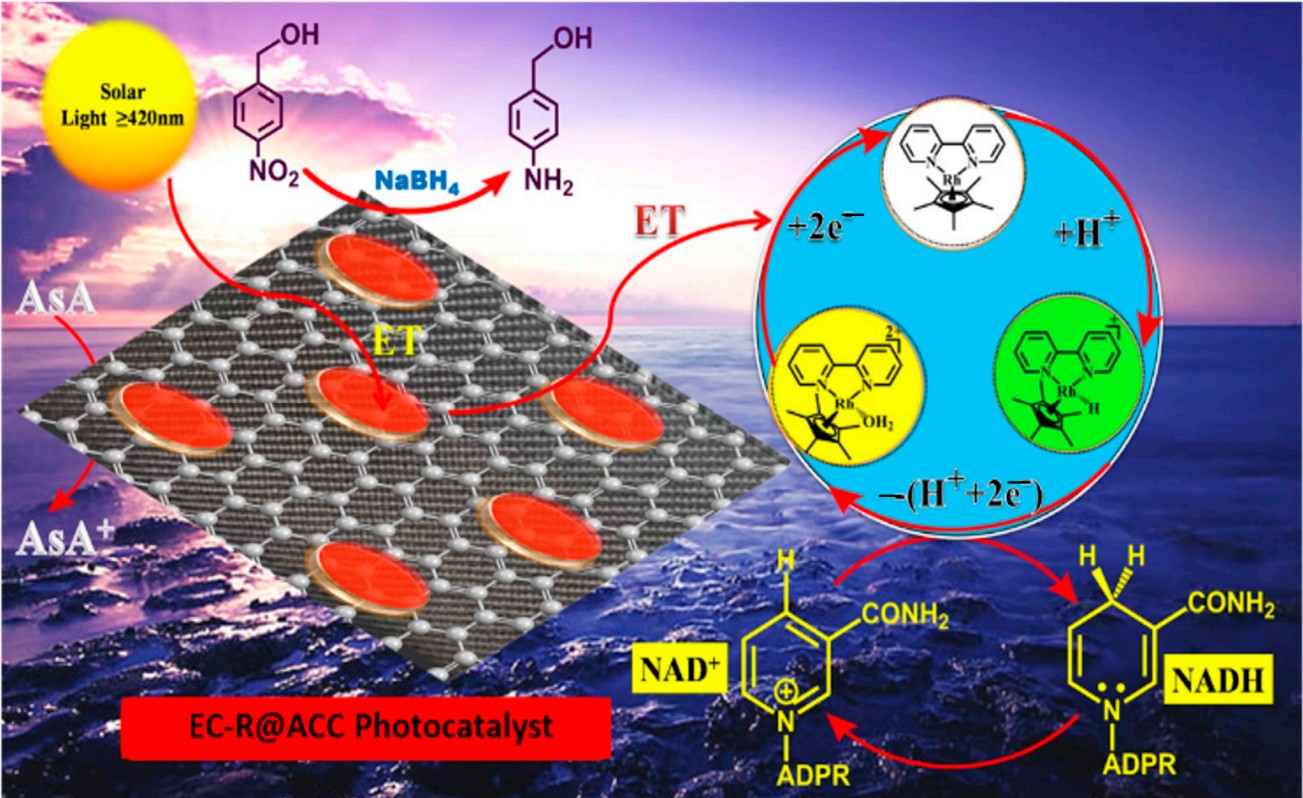

**Scheme 1.** Schematic representation of NADH regeneration and photocatalytic reduction of 4-NBA under solar light.

## 2. Results and Discussion

We introduced our study utilizing 4-NBA (a) as the substrate in the open air. The optimization of the photocatalytic reaction was performed in different solvents (Table 1). For photocatalytic optimization under different solvents, a low-cost, environmentally friendly EC-R@ACC (0.010 g) and 4-NBA (0.045 g) were utilized as a solar light spectrum harvester photocatalyst and starting material model substrate, respectively. The intended 4-ABA product was achieved with 50% conversion when the reaction was optimized in $C_2H_5OH$ (49% yield) in 12 h. We screened the same reaction in different solvents: $C_2H_5OH$, PEG (Polyethylene glycol), $CH_2Cl_2$, and DMF under the same reaction conditions. DMF has the highest polar nature among the solvents, so it provides an excellent yield in 12 h.

We found that when the reaction was carried out with 4-NBA (0.045 g), the EC-R@ACC photocatalyst (0.010 g), and DMF (30 mL) at room temperature under a solar light spectrum for 12 h, the highest conversion (97%) and yield (96%) of 4-ABA was achieved. In contrast, the conversion and yield were reduced in different organic solvents as the reaction time increased, indicating that DMF is the most efficient at promoting the organic transformation reaction [53].

### 2.1. Mechanistic Pathway for the Photoreduction of 4-NBA

Based on the outcomes and the fiction, it is clear that the current photocatalytic reduction pathway includes several critical steps, involving: (I) the adsorption of the reacting molecules on the catalyst's surface, (II) the excitation of the EC-R@ACC photocatalyst to its triplet state and the transfer of electrons easily from sodium borohydride to the newly designed EC-R@ACC photocatalyst, (III) electron transfer from the EC-R@ACC photocatalyst to 4-NBA, (IV) the transfer of hydrogen from the $BH_4$/solvent to 4-NBA, and (V) the desorption of the products from the edges of the newly designed photocatalyst. Scheme 2

depicts the likely mechanistic routes of the current photocatalytic conversion of 4-NBA to 4-ABA.

### 2.2. Presence of Conformational Isomers during 1,4-NADH Synthesis

As shown in Scheme 3, NAD+ is reduced directly for radical coupling and unselective protonation reaction. During this phase, many NAD isomers are formed, which have a presence in both enzymatically inactive and active states. To avoid the generation of enzymatically inactive isomers, an electron mediator must be utilized. Only when exposed to sun rays, the Rh-complex electron mediator supports the creation of the enzymatically solar light spectrum active 1,4-NADH isomer [54].

The reaction buffer medium for NADH regeneration contained 248 µL NAD+ solution, 124 µL Rh-complex, 310 µL ascorbic acid (AsA), 2387 µL phosphate buffer, and 15 mg of the photocatalyst EC-R@ACC photocatalyst. Under continuous solar light irradiation cut by a 420 nm band-pass filter, the reaction was performed in a quartz cuvette as a reactor with a magnetic stir.

**Table 1.** Optimization of photocatalytic reduction of 4-NBA.

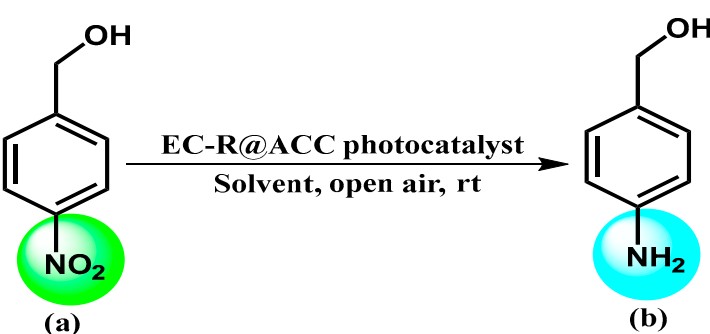

| Entry | Solar Light | Photocatalyst | Solvent | Time | Conversion (%) | Yield (%) |
|---|---|---|---|---|---|---|
| 1. | Yes | EC-R@ACC | $C_2H_5OH$ | 12 | 50 | 49 |
| 2. | Yes | EC-R@ACC | $C_2H_5OH$ | 6 | 47 | 48 |
| 3. | Yes | EC-R@ACC | PEG | 12 | 55 | 56 |
| 4. | Yes | EC-R@ACC | PEG | 6 | 51 | 49 |
| 5. | Yes | EC-R@ACC | $CH_2Cl_2$ | 12 | 60 | 58 |
| 6. | Yes | EC-R@ACC | $CH_2Cl_2$ | 6 | 58 | 57 |
| 7. | Yes | EC-R@ACC | DMF | 12 | 97 | 96 |
| 8. | Yes | EC-R@ACC | DMF | 6 | 79 | 78 |
| 9. | Yes | EC-R | DMF | 6 | 46 | 45 |
| 10. | No | EC-R@ACC | DMF | 12 | 05 | 05 |
| 11. | Yes | EC-R@ACC | Absent | 12 | 10 | 10 |
| 12. | Yes | Absent | DMF | 12 | 05 | 05 |

Reaction conditions: EC-R@ACC photocatalyst (0.010 g), 'a' (0.045 g), and NaBH$_4$ (5 mg/L, various solvents (30 mL)) illuminated under a solar light spectrum for 12 h in an inert atmosphere at room temperature. (a) and (b) represent the reactant and product.

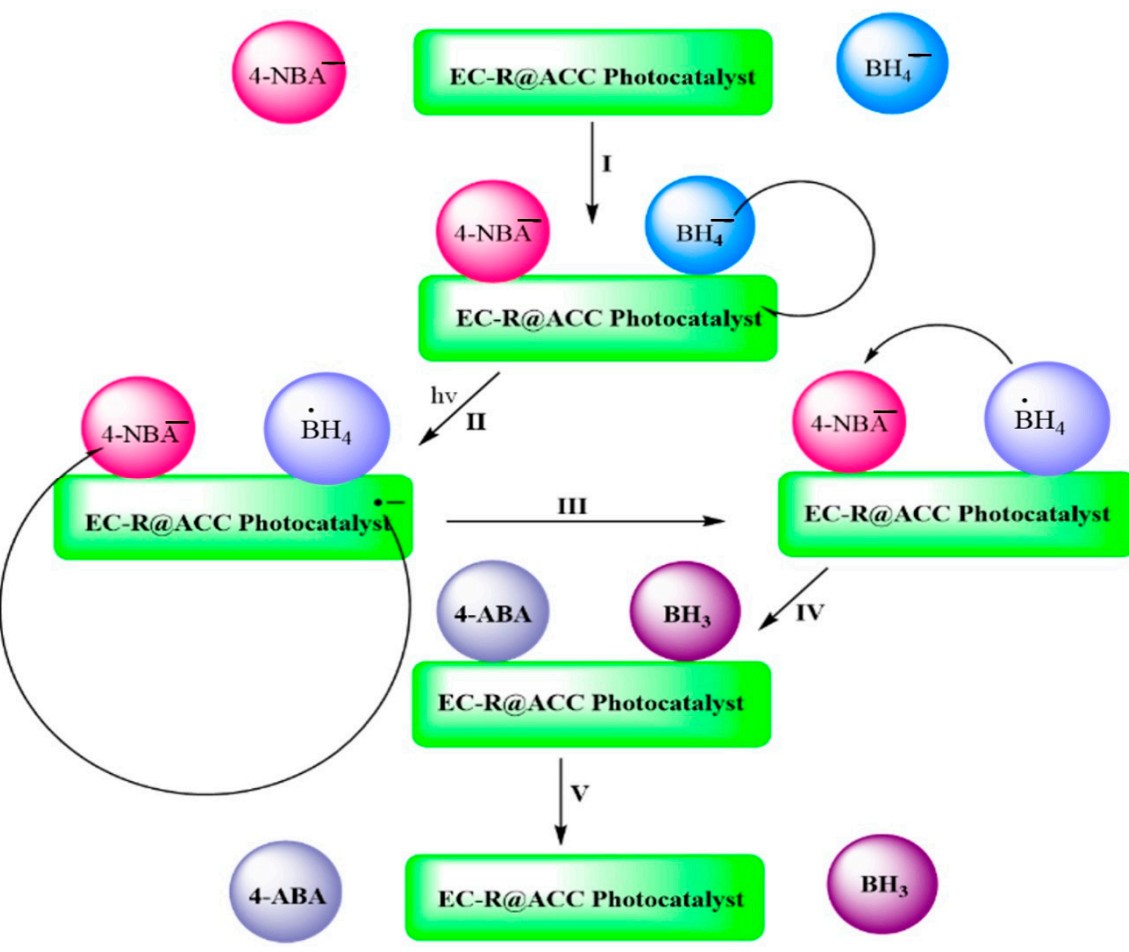

**Scheme 2.** Mechanistic studies of the photocatalytic reduction of 4-NBA to 4-ABA.

The cofactor of 1, 4-NADH rejuvenation was carried out in an inert environment at ambient temperature under the effect of sunlight (>420 nm). An FG@ACC photocatalyst (31 μL), AsA (310 μL), electron mediator (124 μL), and β –NAD$^+$ (248 μL) were dissolved in 2387 μL of sodium phosphate buffer at neutral pH 7.0. First, the process was run in the absence of a solar light spectrum for 30 min, and no cofactor of NADH regeneration was achieved. The cofactor of 1,4-NADH regeneration was achieved in presence of solar light spectrum illumination, as illustrated in Figure 1. It was discovered that as the reaction time increased, so did the yield of 1,4-NADH. The absorbance at 340 nm in the UV-visible spectrum was used to calculate the amount of 1,4-NADH produced. As per a previous report [54], the molar absorption/extinction coefficient of the cofactor of 1,4-NADH is 6.22 mM$^{-1}$ cm$^{-1}$ [54]. We achieved 76.9% catalytic efficiency of 1,4-NADH in two hours (2 h) utilizing a highly stable and solar light active newly designed EC-R@ACC photocatalyst (shown in Figure 1). Because of the π-π interaction, the photocatalytic ability of the solar light active newly designed EC-R@ACC photocatalyst is greater than that of its precursor EC-R [40,54].

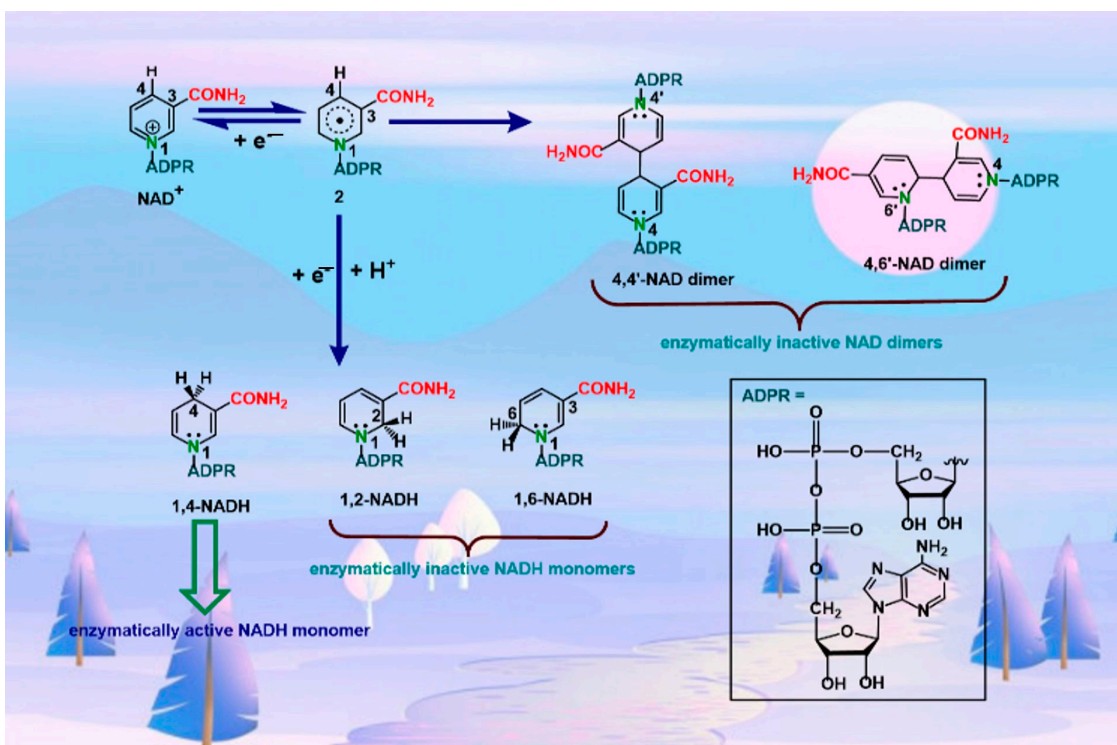

**Scheme 3.** The enzymatically active 1,4 NADH cofactor regeneration after NAD+ reduction.

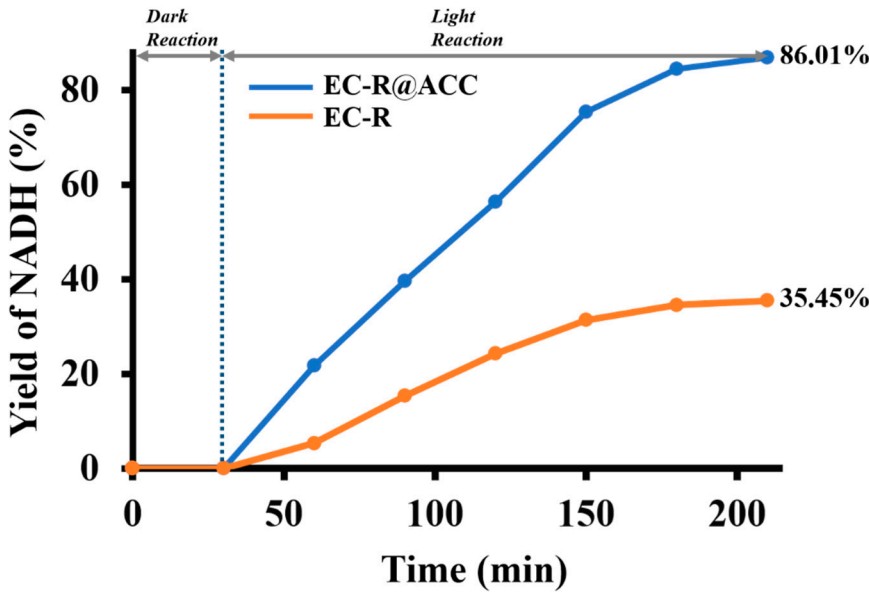

**Figure 1.** The photocatalytic activity of the EC-R@ACC photocatalyst and EC-R for NADH regeneration under solar light.

### 2.3. Reaction Mechanism of Photocatalytic NADH Regeneration

Scheme 4 demonstrates a possible approach for photocatalytic regeneration of NADH using the EC-R@ACC photocatalyst. During the photocatalytic activity, the cationic type of the electron mediator (A) hydrolyzes, yielding a water-coordinated complex (B), symbolized as $[Cp*Rh(bpy)(H_2O)]^{2+}$. The complex (B) interacts with the formate ($HCOO^-$) during the hydride removal procedure [54]. This reaction generates the Rh hydride complex (C), i.e., $[Cp*Rh(bpy)(H)]^+$, and the release of $CO_2$. When the EC-R@ACC photocatalyst contributes electrons to the complex of Rh, the reduced intermediate complex (D) is generated

(C). NAD$^+$ interacts with complex D via the activity of amide and transfer of hydride, due to which the NADH cofactor's region-selectivity regenerates.

**Scheme 4.** Photocatalytic NADH regeneration utilizing an EC-R@ACC photocatalyst mechanism.

### 2.4. Solar Light-Induced Catalytic 1,4 NADH Regeneration

We exclusively focused on regenerating the enzymatically active form of 1,4-NADH from the oxidized form of NAD$^+$. As shown in Scheme 5, an electron mediator was used to prevent the transformation of undesirable isomeric forms, resulting in the artificial photocatalytic transformation of 1,4 NADH under sunlight irradiation. A neutral solution (pH 7.0) of phosphate-buffered solution (NaH$_2$PO$_4$–Na$_2$HPO$_4$, 0.1 M) and an NAD+ cofactor along with scavenger agents were used to regenerate NADH. In addition, combined with the recently synthesized EC-R@ACC photocatalyst, [Cp*Rh(bpy)Cl]Cl was introduced to the reaction media.

Recycling experiments for NADH synthesis in the presence of EC-R@ACC photocatalyst were conducted by recycling the newly designed same photocatalyst several times under identical conditions to investigate the utility and sustainability of the EC-R@ACC photocatalyst under the same experimental conditions. During the reusability test, a nearly constant conversion yield was observed with no appreciable decline in efficiency, suggesting that the EC-R@ACC photocatalyst has strong catalytic strength [55]. Furthermore, the extra experiments were carried out under sunlight in the absence of NAD$^+$. No absorbance peak was observed at a wavelength of 340 nm in this experiment, which suggests that NADH cannot be produced in the absence of NAD$^+$ (Figure 1). Generally, every component of the artificial photosynthetic machinery is important, including solar light, EC-R@ACC,

and NAD⁺. It should be noted that to eliminate the photo-saturation during the measurement of UV-Visible spectra, the concentration of the reaction media, which included AsA, NAD⁺, EC-R@ACC, and Rhodium complex, was kept quite low [54,55].

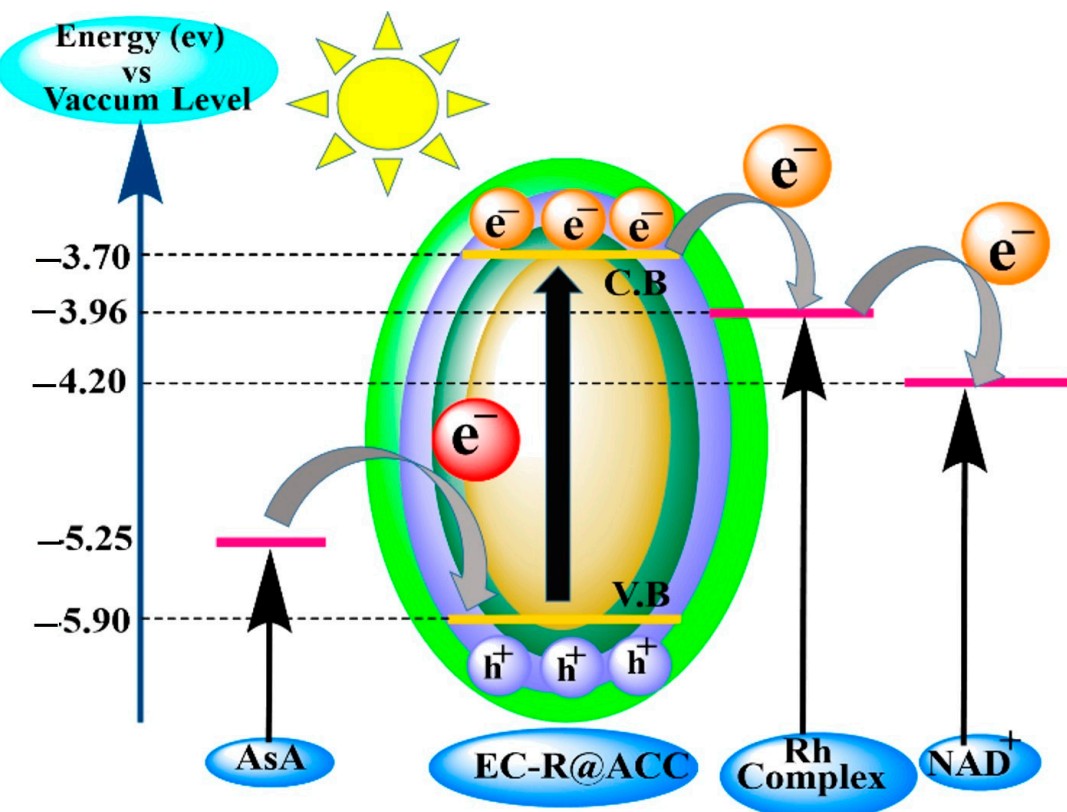

**Scheme 5.** A simplified potential energy diagram showing carrier generation and its migration in the photocatalytic system.

Scheme 5 shows the energy-labeled diagram, which depicts the pathway of the induced charge carriers in the photocatalytic system. Initially, on the irradiation of solar light, ascorbic acid becomes oxidized, and the electron of the EC-R@ACC photocatalyst is transferred from the valence band/highest occupied molecular orbital (HOMO) to the conduction band. Subsequently, the electron jumps from the conduction band/lowest unoccupied molecular orbital (LUMO) of the EC-R@ACC photocatalyst to the conduction band Rh-complex due to the lower band gap of the Rh-complex. Thus, the Rh-complex acts as an electron mediator. After the Rh-complex electron, the electron follows the same manner and easily jumps to the conduction band of NAD⁺. Here, NAD⁺ is reduced in NADH (Nicotinamide Dinucleotide Adenine Hydrate). In this photocatalytic process, after the reduction of NADH, the photocatalytic system is able to follow the Calvin cycle and mimic the natural photosynthetic route [54].

### 2.5. Study of UV-Visible Spectra of Newly Designed Solar Light Spectrum Responsive EC-R@ACC Photocatalyst

UV-Visible spectroscopy (UV-Visible-1900i, Shimadzu, Japan) was utilized to study the absorption spectrum of the ACC and EC-R@ACC in DMF (Figure 2). The UV-Visible spectra of the ACC were observed at about 250 nm [56], whereas the absorption band of the EC-R@ACC was observed at 545 nm. We estimated the optical band gap using the Scherrer equation (1240/λ) and found it to be 4.96 eV and 2.29 eV, respectively, indicating that it can operate as an active catalyst. The predicted optical band gap (2.29 eV) validates redshift and boosts solar-driven activation. The results showed that the EC-R and ACC absorb a lot of visible light. The absorption spectra of the newly designed EC-R@ACC

photocatalyst in the bathochromic shift are most significant and are responsible for cofactor 1,4-NADH regeneration and organic transformation, which improves its solar light harvesting abilities/capabilities in the solar light spectrum region.

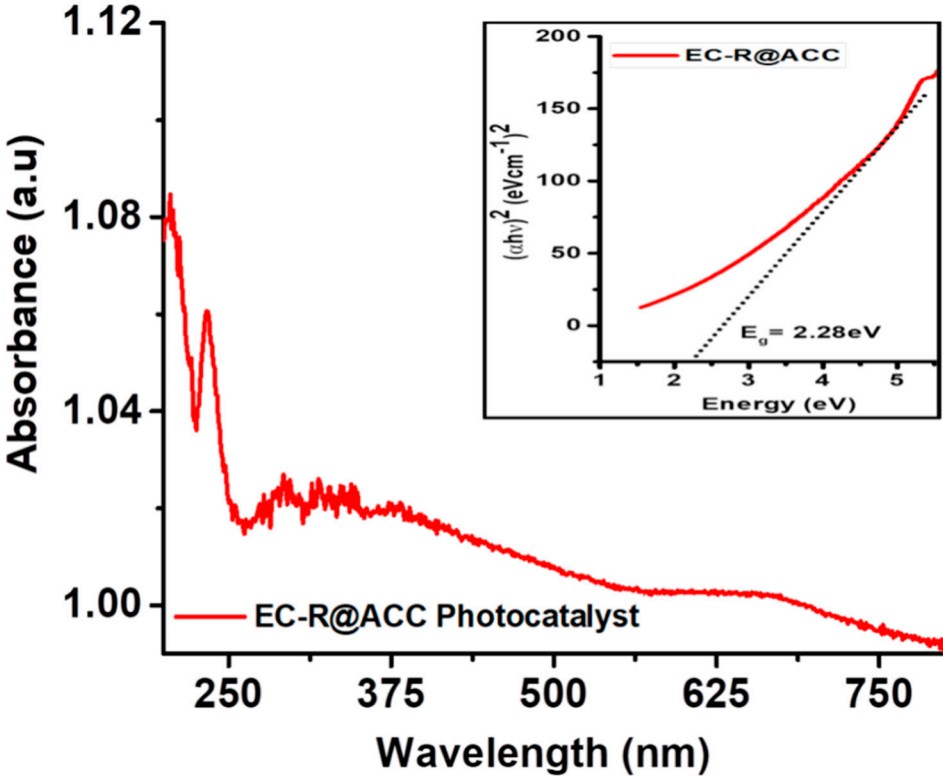

**Figure 2.** UV-Visible absorption spectra of the photocatalyst.

The optical band gap of the EC-R@ACC photocatalyst was computed using the Scherrer equation (1240/λ) [57], and it is close to 2.28 eV at about 540 nm. The cyclic voltammetry (CV, K-lyte electrochemical station,) measurement supports the computed optical band gap by the Scherrer method with the value of 2.20 eV [57]. The reduction and oxidation energy potential values of the newly designed EC-R@ACC photocatalyst were achieved by the CV measurement technique (Figure 3). The EC-R@ACC photocatalyst oxidation and reduction potential values were +1.10 V and −1.10 V, respectively. The collected reduction and oxidation energy potential data were utilized to calculate the energy gap/band gap.

The CV experiment authorizes the energy gap/band gap calculation (see Figures 3 and 4) [57]. A CV experiment was used to measure the reduction and oxidation energy potential values of the EC-R@ACC photocatalyst. The reduction and oxidation energy potentials were measured as +1.10 V and −1.10 V, respectively. The reduction and oxidation values gathered can be utilized to compute the band gap using the Latimer diagram (Figure 4), which verifies the optical band gap. Bathochromic shifts in the absorption spectra of the EC-R@ACC photocatalyst were detected, which boosts its ability to harvest sunlight.

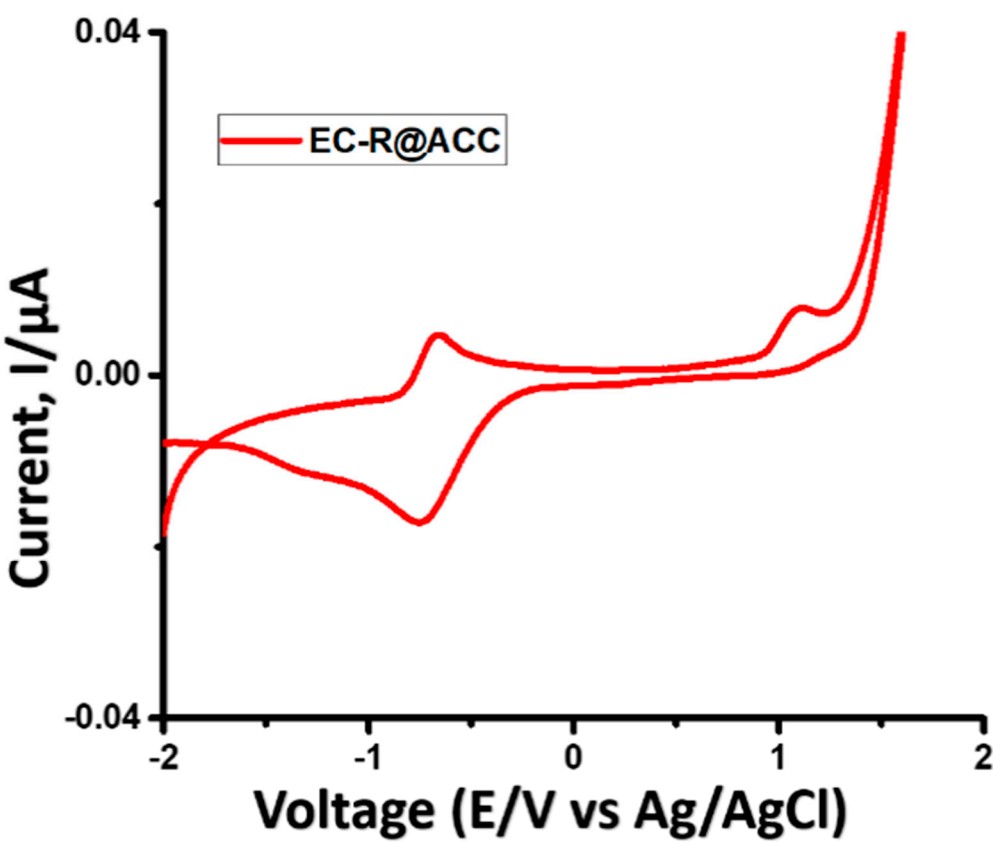

**Figure 3.** Cyclic voltammetry of the EC-R@ACC photocatalyst.

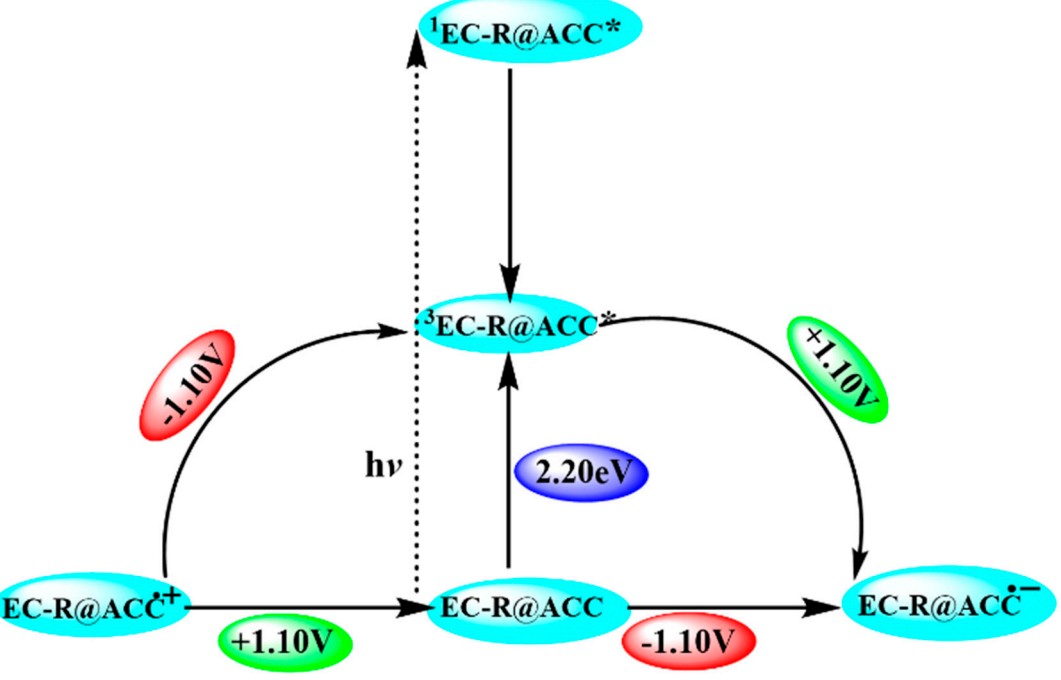

**Figure 4.** Latimer Diagram displaying the photo-redox property of the EC-R@ACC photocatalyst; (* represent the excited state.)

### 2.6. Study of Zeta Potential of ACC and EC-R@ACC Photocatalyst

The zeta potential (Malvern Panalytical, Nano-zetasizer (NZS90), Malvern, UK) of the EC-R@ACC photocatalyst was observed and showed an additional negative value of

−40.1 mV, while the ACC showed a value of −23.7 mV (Figure 5) [58]. It is illustrated that the synthesis of the EC-R@ACC composite provides the more negatively charged fractions as it has a high content of EC-R. Additionally, the more negative zeta potential value for the EC-R@ACC photocatalyst proves that the interaction between the ACC and EC-R is quite good [59].

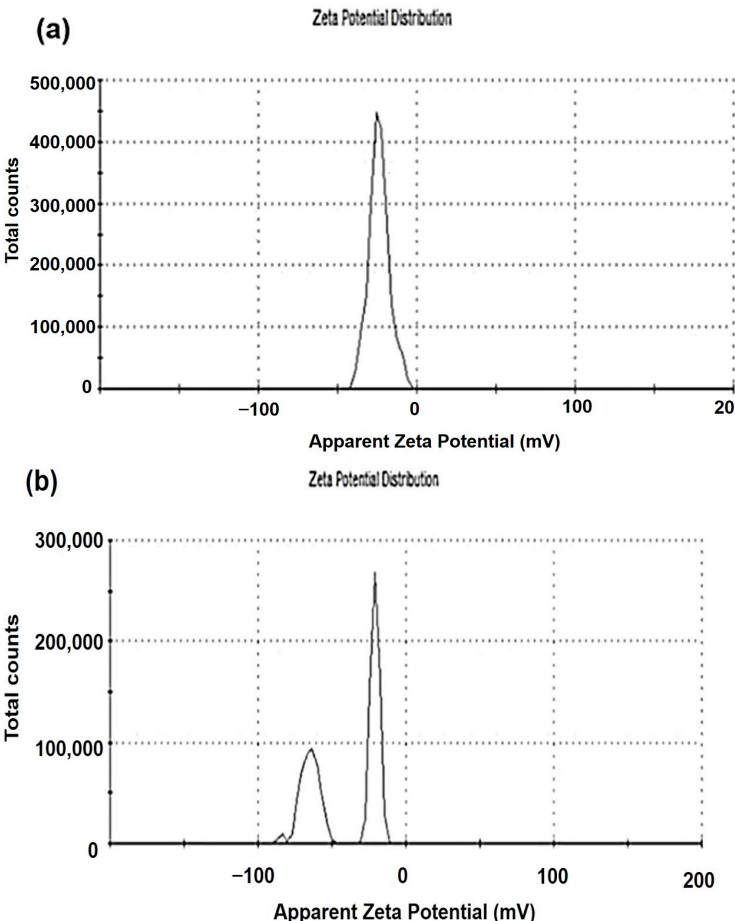

**Figure 5.** Studies of the (**a**) ACC (−23.7 mV) and (**b**) EC-R@ACC solar light spectrum photocatalyst (−40.1 mV) by Zeta potential pathway.

The FTIR spectra (Shimadzu, IRspirit FTIR-8000, Anan, Japan) of the EC-R@ACC photocatalyst, as well as the ACC and EC-R in Figure 6, demonstrated the occurrence of an interaction in the EC-R@ACC. Figure 6 indicates that the FTIR spectrum of the EC-R@ACC displays a stretching peak of approximately 3450 cm$^{-1}$, confirming the existence of the -OH group [60]. The stretching peak of $SO_3^-$ is also found at about 1250 cm$^{-1}$ [61]. The stretching peak of –COONa is also found at about 950 cm$^{-1}$ [62]. The stretching peak of –CO is also found at about 1050 cm$^{-1}$ [40]. The stretching peak of –CH$_3$ is also found at about 2850 cm$^{-1}$; however, it is completely absent in the ACC FTIR spectra [61]. The results show that the interaction in the EC-R@ACC photocatalyst was formed successfully. Additionally, we recycled the EC-R@ACC photocatalyst for more than four consecutive runs (i.e., four reuses) under the same reaction circumstances (Figure 7). It was perceived that the photocatalytic cofactor 1,4-NADH regeneration is almost constant in all the re-cycles, confirming the highest solar light spectrum harvesting stability of the EC-R and EC-R@ACC photocatalyst, respectively. In addition, the observed results revealed that the EC-R@ACC photocatalyst possesses higher stability (Figure 7a) compared to the EC-R pho-tocatalyst (Figure 7b), which clearly revealed that the EC-R@ACC is superior to the EC-R

photocatalyst. The turnover number (TON) of the EC-R photocatalyst for the reduction of 4-nitrobenzyl alcohol is calculated from the below-mentioned equation [63]:

TON = No. of substrate molecules converted into the product by 1 g of photocatalyst

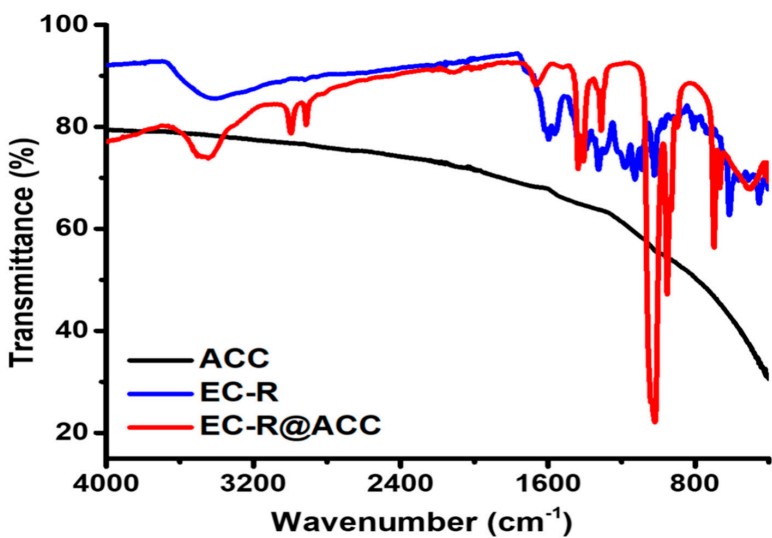

**Figure 6.** Studies of the ACC, EC-R, and EC-R@ACC photocatalyst by FTIR technique.

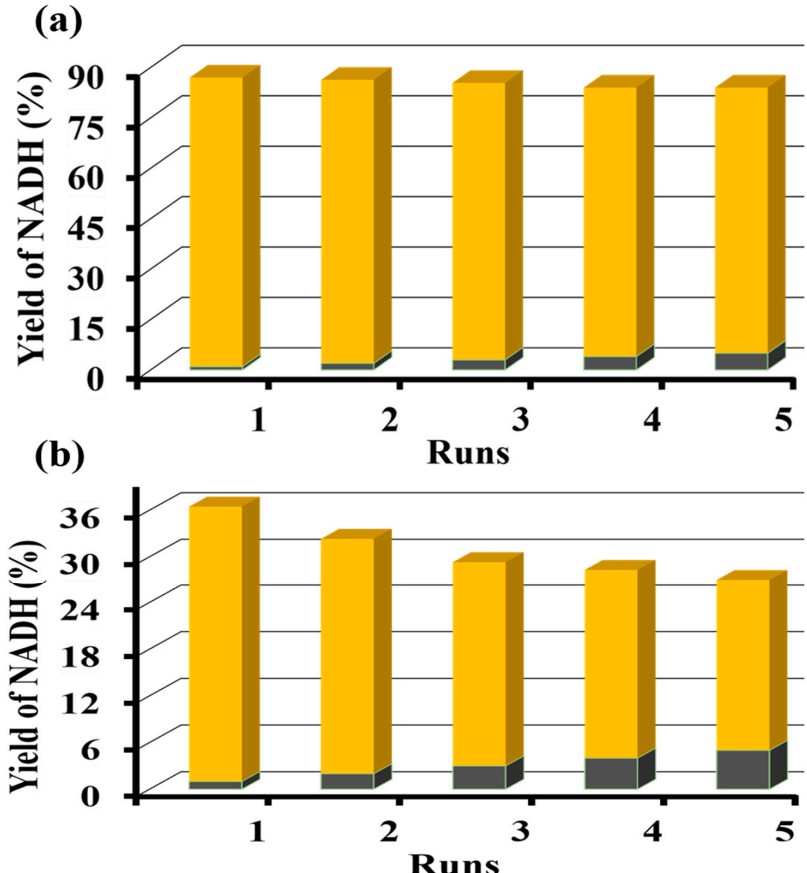

**Figure 7.** The recycle stability for NADH regeneration by the (**a**) EC-R@ACC photocatalyst and (**b**) EC-R photocatalyst.

So, the calculated TON of the EC-R@ACC photocatalyst for the reduction of 4-nitrobenzyl alcohol is $1.769 \times 10^{19}$ molecules.

## 3. Experimental Details

### 3.1. Materials and Chemicals

The ACC, sodium borohydride ($NaBH_4$), EC-R, N, N-dimethyl formamide (DMF), 4-nitro benzyl alcohol (4-NBA), sodium phosphate monobasic dihydrate ($NaH_2PO_4 \cdot 2H_2O$), sodium phosphate dibasic dihydrate ($Na_2HPO_4 \cdot 2H_2O$), nicotinamide adenine dinucleotide ($NAD^+$), and 2,2 bipyridine (pentamethylcyclopentadienyl) rhodium (III) chloride dimer were purchased from Sigma Aldrich (Munich, Germany) and TCI (Portland, OR, USA).

### 3.2. Synthesis of ACC

Activated Carbon Cloth (ACC) was synthesized in the reported way. Carbon fabric (1 cm × 1 cm) was initially washed many times with acetone and distilled water. Following multiple washes, the carbon fabric was cured with conc. $HNO_3$ at more than 90 °C for roughly 4 h. Following the acid treatment, the carbon fabric was thoroughly cleaned with distilled water and acetone. After washing, the freshly produced activated carbon cloth (ACC) was dried in a 70 °C oven [64].

### 3.3. Synthesis of EC-R@ACC Photocatalyst

Typically, 350 mg of carbon powder (graphene) and 150 mg of EC-R were mixed in 20 mL DMF and stirred for 2 h to ensure complete mixing. Then, the solution was autoclaved at 150 °C for 12 h (Figure 8). Furthermore, the solution was cooled to room temperature. Then, the solvent in the solution was evaporated at its boiling point. The obtained compound was thoroughly washed with distilled water 2–3 times. Finally, the newly designed EC-R@ACC photocatalyst was dried in the oven overnight at 100 °C. The amount of EC-R@ACC achieved was 203 mg [65].

### 3.4. Synthesis of Rh-Complex

The Rh-complex $[Cp*Rh(bpy)Cl]^+$ was prepared using a well-standard technique. In 5 mL distilled methanol, 0.025 g of rhodium compound ($[Rh(C_5Me_5)Cl_2]_2$) was dissolved in an $N_2$-purged environment. The methanol solution was then mixed in a dark-incubated environment at room temperature with 0.013 g of 2,2′-bipyridyl (2 eq.) [15].

As soon as diethyl ether was added, a yellow precipitate formed. In an $N_2$-purged environment, the complete product was received by the filtration method and dried at room temperature.

### 3.5. Synthesis of 4-ABA

The mixture of the EC-R@ACC photocatalyst (0.010 g), 4-NBA (0.045 g), and $NaBH_4$ (5 mg/L) was prepared in 30 mL DMF in a glass vial and mixed with a magnetic stirrer. The reaction mixture was stirred at room temperature for 12 h in the presence of air under continuous high solar irradiation. The reaction mixture was then examined using TLC after it was completed. After filtering, the mixture was thoroughly washed with 50 mL of distilled water. The filtrate was concentrated using a rotary evaporator to abstract the final product. The compound's yield was 97.61% [52].

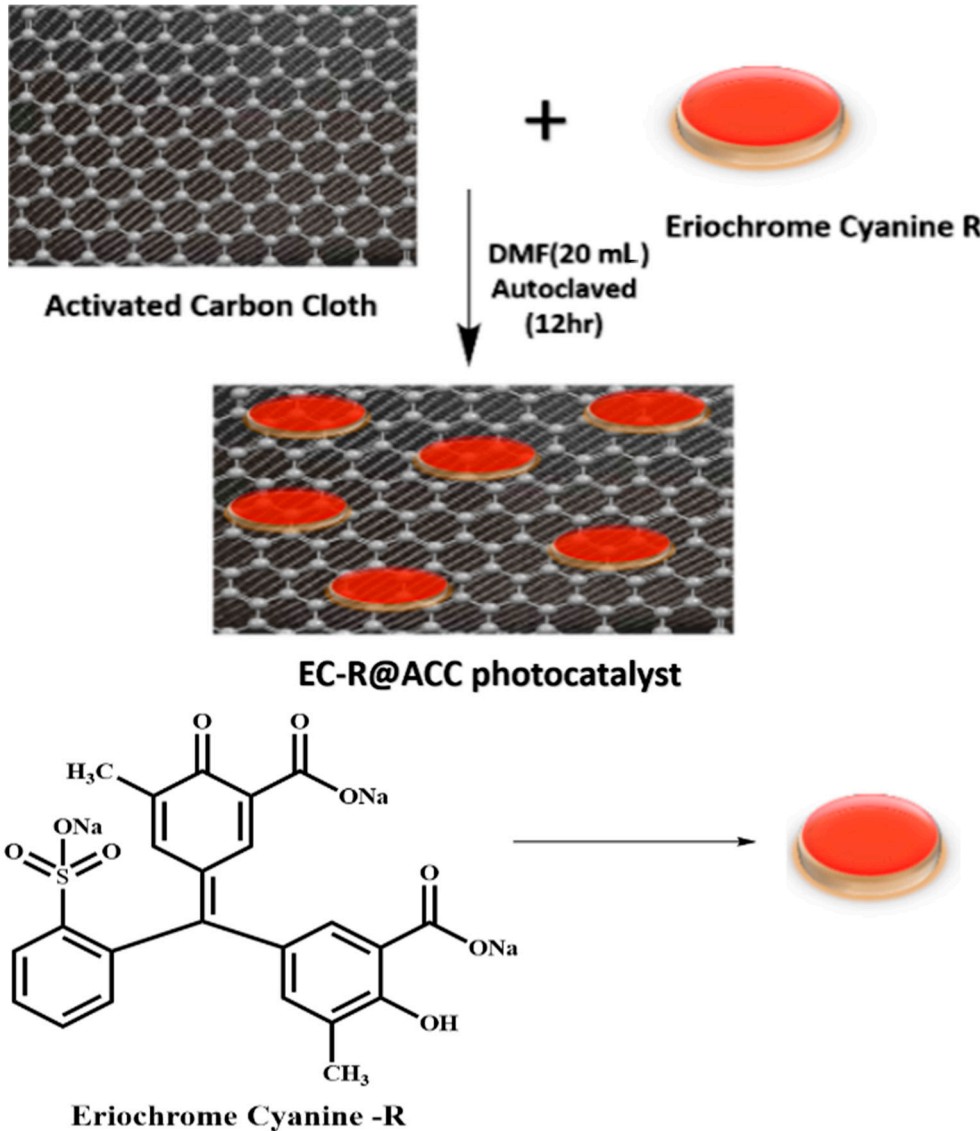

**Figure 8.** The schematic diagram for the synthesis of the EC-R@ACC photocatalyst.

## 4. Conclusions

Overall, with this support, we have explained that the newly designed photocatalyst is feasible for the regeneration of NADH and organic transformation under solar light. In this context, photochemically under solar light irradiation, the regeneration of NADH and the reduction of 4-nitrophenol with $NaBH_4$ can be carried out using a metal-free ACC templated EC-R@ACC photocatalyst. The regeneration of NADH, as well as the reduction of 4-NBA into 4-ABA, was accomplished using an EC-R doped ACC photocatalyst (EC-R@ACC) in conjunction with artificial photosynthetic machinery. The EC-R@ACC photocatalyst demonstrated good maintenance of catalytic effectiveness during numerous cycles of photocatalytic reaction due to its great thermal and chemical stability. Most importantly, the EC-R@ACC, under continuous solar light irradiation, permits the effective regeneration of NADH cofactors with a yield of 76.9%. This research suggests that solar light could be used to produce more effective and cost-effective NADH regeneration along with photocatalytic reduction of 4-NBA and many more reductive processes.

**Author Contributions:** Conceptualization, V.G., R.K.Y., A.U. and R.K.S.; software, V.G., R.K.Y., A.U., A.A.I., S.S., R.S., R.K.S., D.T., D.K.D., A.K.S. (Alok Kumar Singh), A.K.S. (Atresh Kumar Singh) and S.B., validation, V.G., R.K.Y., A.U., A.A.I., S.S., R.S., R.K.S., D.T., D.K.D., A.K.S. (Alok Kumar Singh), A.K.S. (Atresh Kumar Singh) and S.B.; formal analysis and investigation, V.G., R.K.Y., A.U., A.A.I., S.S., R.S., R.K.S., D.T., D.K.D., A.K.S. (Alok Kumar Singh), A.K.S. (Atresh Kumar Singh) and S.B.; writing—original draft, V.G., R.K.Y., A.U. and R.K.S.; writing—review and editing, V.G., R.K.Y., A.U., A.A.I., S.S., R.S., R.K.S., D.T., D.K.D., A.K.S. (Alok Kumar Singh), A.K.S. (Atresh Kumar Singh) and S.B.; visualization, D.K.D. and S.B.; project administration, R.K.Y. All authors have read and agreed to the published version of the manuscript.

**Funding:** The authors are thankful to the Deanship of Scientific Research and supervision of the Centre for Scientific and Engineering Research at Najran University, Najran, Kingdom of Saudi Arabia for funding under the Research Centers funding program Grant No. NU/RCP/SERC/12/6.

**Data Availability Statement:** Not applicable.

**Acknowledgments:** The authors are thankful to the Deanship of Scientific Research and supervision of the Centre for Scientific and Engineering Research at Najran University, Najran, Kingdom of Saudi Arabia for funding under the Research Centers funding program Grant No. NU/RCP/SERC/12/6.

**Conflicts of Interest:** The authors declare no conflict of interest.

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
