# Peer review of "Highly Efficient Self-Assembled Activated Carbon Cloth-Templated Photocatalyst for NADH Regeneration and Photocatalytic Reduction of 4-Nitro Benzyl Alcohol"

_catalysts, doi:10.3390/catal13040666_

Round 1

Reviewer 1 Report

This manuscript synthesis of self-assembled activated carbon cloth-Templated photocatalyst for NADH regeneration and photocatalytic reduction of 4-nitro benzyl alcohol. Several issues should be addressed before publication as follow:

1. In Figure 1, the photocatalytic activity of EC-R@ACC photocatalyst and ACC for NADH 136 regeneration under solar light should be measured again. From this data, it can be observed that the photocatalytic activity increases with the reaction time. In this reagard, what about the one when increasing reaction time to 180 min or 300min? Is it possible to achieve 100% activity?

2.  The cycle photocatalytic activity of EC-R@ACC photocatalyst and ACC for NADH 136 regeneration should be provided. At the same time, what about the stability of the photocatalysts?

3. Figure 3 needs to test again. Please provide the data of the wavelength from 200 to 800 nm.

4. Figure 7 is deformed.

5. The references style should be the same.

Reviewer 2 Report

This paper "Highly Efficient Self-Assembled Activated Carbon Cloth-Tem-2 plated Photocatalyst for NADH Regeneration and Photocatalytic Reduction of 4-Nitro Benzyl Alcohol" is at a draft stage.

Both abstract and introduction include generality with little relation with the core of the paper.

Some statements as "4-nitrobenzyl alcohol is one of the most common by-products that is harmful to the environment" are stated without references.

The objective of the paper is a "greener" synthesis method for that same compound but the experimental concentration of 1.5 g/L in DMF with a huge excess of NaBH4 make this procedure (however interesting) impossible to industrialise !

The data in table 1 are given with a outstanding precision of 0.01% without explanation of the analysis method ! Moreover  the terms Yield and Selectivity are used wrongly (I guess it is Conversion and Yield respectively).

The link between the main reaction and the regeneration method is lacking and it's hard to understand the relation between this 2 parts of the article.

In the conclusion is claimed "numerous cycles of photocatalytic reaction" without any provided data to support this claim.

Scheme and graphic quality is very low.

Round 2

Reviewer 1 Report

It is recommended for rpublication in present form.

Reviewer 2 Report

I thank the authors for having considered my comments.

* Concerning the concentration of NaBH4 (table 1) the corrected value is around half the stoechiometry thus scheme 2 should be adapted to add reduction using BH3. And is it 5mg/30 ml (table 2) or 5mg/L (chapter 3.5) ???

* Concerning the figure 8, it is not clear what are the differences between 8a and 8b(yield of NaDh in both case). Moreover 30% deactivation is noticed after 5 runs, it's definitively not the "highest stability" and a TON calculation whould be appreciated.
